# Difference in Correction Power between Hybrid Lateral Closed-Wedge High Tibial Osteotomy and Medial Open-Wedge High Tibial Osteotomy was Associated with Severity of Varus Deformity and Different Hinge Distance from Center of Deformity

**DOI:** 10.3390/diagnostics14111137

**Published:** 2024-05-29

**Authors:** Seok Jin Jung, Jun Ho Kang, Seung Joon Rhee, Sang Won Moon, Lih Wang, Darryl D D’Lima

**Affiliations:** 1Department of Orthopedic Surgery, Biomedical Research Institute, Pusan National University Hospital, Busan 49241, Republic of Korea; 2College of Medicine, Pusan National University, Busan 46241, Republic of Korea; 3Department of Orthopedic Surgery, Inje University Haeundae Paik Hospital, Busan 48108, Republic of Korea; 4Department of Orthopedic Surgery, College of Medicine, Dong-A University, Busan 49201, Republic of Korea; 5Department of Molecular Medicine, Scripps Research, La Jolla, CA 92037, USA; 6Shiley Center for Orthopaedic Research and Education at Scripps Clinic, Scripps Health, La Jolla, CA 92121, USA

**Keywords:** osteotomy, knee, tibia, osteoarthritis, retrospective study

## Abstract

Hybrid lateral closed-wedge high tibial osteotomy (HBHTO) carries certain advantages over medial open-wedge high tibial osteotomy (OWHTO). We investigated the potential difference in the required correction angle between HBHTO and OWHTO to achieve an equal amount of whole lower-extremity alignment correction, retrospectively analyzing the preoperative plain radiographic images of 100 patients. The medial proximal tibial angle (MPTA), joint line convergence angle (JLCA), mechanical lateral distal femoral angle (mLDFA), hip–knee–ankle axis (HKA), length of the tibia, width of the tibial plateau, length of the lower limb (leg length), and location of the center of deformity (CD) were measured. Differences in the required correction angle at the hinge point between the two techniques (CAD) were compared, and correlation analysis was performed to reveal the influential factors. The mean difference in CAD between HBHTO and OWHTO was 0.78 ± 0.22 (0.4~1.5)°, and mean WBL position change per correction angle was 3.9 ± 0.3 (3.0~4.6)% in HBHTO and 4.1 ± 0.3 (3.1~4.7)% in OWHTO. Correlation analysis revealed a strong positive correlation between CAD and HKA. mLDFA, JLCA, MPTA, leg length, OWCD, HBCD, and HCD were also significantly correlated with CAD. HBHTO required a 5.6% larger correction angle at the hinge point to achieve the same amount of alignment correction as OWHTO.

## 1. Introduction

Valgus producing high tibial osteotomy (HTO) is a standard treatment option for medial unicompartmental osteoarthritis (OA) of the knee with varus deformity. Numerous studies have reported reproducible and reliable mid-to-long-term HTO results [1,2,3,4]. There are two representative surgical options in HTO. Most surgeons have performed medial open-wedge HTO (OWHTO) since an angular stable locking plate was developed in the 1990s [5]. The popularity of OWHTO is due to its relatively brief surgical technique and safety from peroneal nerve injury compared to conventional lateral closed-wedge HTO (CWHTO). In OWHTO, only one coronal-plane osteotomy is required to form the main construct, and the correction angle can be easily adjusted by distracting the osteotomy gap around the lateral hinge. However, meticulous surgical techniques are required to form an intact hinge point, which is an essential element of the biomechanical stability of this technique. Nonunion or delayed union of the osteotomy gap is common if a hinge fracture occurs or angular stable fixation is not achieved.

Recently, CWHTO has attracted the interest of surgeons owing to the hybrid lateral closed-wedge high tibial osteotomy (HBHTO) technique [6], which has considerably improved the shortcomings of conventional CWHTO. Lateral cortical step-off, which is disadvantageous for optimizing firm fixation and early weight bearing in CWHTO [1], could be minimized by an obliquely oriented osteotomy line in HBHTO. Moreover, large angular correction could be performed in HBHTO with less concern regarding the amount of bone loss and tibia length change owing to the laterally located hinge point relative to the conventional CWHTO and fewer concerns regarding hinge fracture powered by the completely separating hinge point, which is quite different from both OWHTO and CWHTO [7]. Faster osteotomy site union is the foremost advantage of HBHTO over OWHTO [8].

Despite the fascinating advantages of the hybrid technique, we recognize that there might be some hidden trade-offs between the advantages and disadvantages of HBHTO and OWHTO. We attempted to investigate the potential difference in the required correction angle between HBHTO and OWHTO to achieve an equal amount of whole lower-extremity alignment correction and to delineate the factors that influence this difference.

We hypothesized that the amount of angular correction in the proximal tibia to achieve an equal amount of whole lower-extremity alignment correction would differ between HBHTO and OWHTO. We also hypothesized that the difference in the correction angle originates from the difference in the distance between the hinge and center of the deformity in each osteotomy technique.

## 2. Materials and Methods

We performed a retrospective analysis of patients who visited our institute to treat medial compartmental OA between 2014 and 2022. Patients with varus lower-limb alignment and medial compartment OA were included in this study. Patients with a varus alignment exceeding 20° or valgus alignment were excluded. One hundred patients were finally included and their plain radiographic images were analyzed. The medial proximal tibial angle (MPTA), joint line convergence angle (JLCA), mechanical lateral distal femoral angle (mLDFA), and hip–knee–ankle axis (HKA) were measured in the weight-bearing long-standing anteroposterior view of the whole lower limb (orthoroentgenogram) using the AI-based roentgenographic image reading software Connecteve-X (Connecteve, Seoul, Republic of Korea) (Figure 1).

The length of the tibia, the width of the tibial plateau, the length of the whole lower limb from the femoral head center to the tibial plafond center (leg length), and the weight-bearing line passing point were measured on the orthoroentgenogram by two board-certified orthopedic surgeons and one radiologist with over 10 years of experience in their specialty. A design for virtual HTO using the ‘Miniaci’ method [9] was performed on all the patients’ orthoroentgenograms in two settings. The first design was an OWHTO with osteotomy (OT) starting 45 mm below the medial cortical edge of the medial tibial plateau and ending at the lateral hinge point, which was located 10 mm inferior to the lateral tibial plateau and 10 mm medial to the lateral tibial cortex. The second design was HBHTO, with the OT starting 35 mm below the lateral cortical edge of the lateral tibial plateau and ending 15 mm below the tibial plateau’s medial cortical edge. The hinge of the HBHTO was located on the OT line at two-thirds of the lateral tibial cortex and one-third of the medial tibial cortex. Fibular shortening was considered for a method to enable closing of the main osteotomy gap. The target alignment was set as the weight-bearing line (WBL) passing through the tibial plateau 65% from the medial cortical edge (Figure 2). 

The correction angle around the hinge point, which is in the proximal tibia, required to achieve the same amount of WBL change could be calculated using the two design methods. The correction angles were compared between the two settings, and a correlation analysis was performed to delineate the factors influencing the difference between the correction angles. The difference in the required correction angle between HBHTO and OWHTO was defined as the correction angle difference (CAD). The center of deformity (CD) was defined as a point formed by a crossing point between the mechanical axis of the femur and tibia. The distances between the hinge point and CD were measured for HBHTO (HBCD) and OWHTO (OWCD). Each parameter was measured twice by each rater, and the measurements were compared for intra- and inter-rater agreement. We divided the patients into two groups according to the position of CD in the tibia. CD position in the proximal one-fourth of the tibia was defined as middle CD position group and CD distal to the proximal one-fourth was defined as low CD position group. Radiological parameters were compared between the two groups. This study was approved by the Institutional Review Board of our hospital (2308-016-130). Informed consent was obtained from all patients.

### Statistical Analysis

All statistical analyses were performed using SPSS software (version 27.0; IBM Corp., Armonk, NY, USA). Continuous variables were compared as the means ± standard deviations using a 2-sample Student’s *t*-test or paired *t*-test test. Correlation analysis between CAD and various factors was performed using Pearson’s correlation coefficient, and the influence of these factors on CAD was delineated using multiple regression analysis. Inter- and intraobserver reliabilities were assessed using the intraclass correlation coefficient (ICC).

## 3. Results

Orthoroentgenograms of 24 men and 76 women with a mean age of 63.8 (39–85) years were assessed. Mean HKA was 9.94 ± 3.32 (3.7–17.6)° with mLDFA 89.03 ± 2.23 (83.7–97.9)°, MPTA 82.49 ± 2.07 (75.8–88.4)°, and JLCA 4.64 ± 2.93 (–3.4–12.6)°. The mean leg length was 715.18 ± 46.05 (611.1–838.17) mm with a tibia length of 309.91 ± 30.15 (103.8–378.2) mm and proximal tibial width of 72.88 ± 5.11 (64.58–86.38) mm. The mean WBL position was 7.4 ± 13.3(−29.4–31.2)% from the medial tibial plateau edge (Table 1).

The ICCs for inter- and intra-rater reliabilities were all >0.8, ranging from 0.818 to 0.991, for all human radiological parameters, indicating good reliability (Table 2).

The mean required correction angle at the hinge point in the proximal tibia to achieve 65% WBL was 14.86 ± 3.47 (8.3–24)° in the HBHTO design and 14.09 ± 3.29 (7.9–22.9)° in the OWHTO design, which was significantly different in each of the 100 cases (*p* < 0.001). The mean difference in the required correction angles between the two designs (CAD) was 0.78 ± 0.22 (0.4–1.5)° (Figure 3).

The mean ratio of the required correction angle in HBHTO relative to OWHTO was 1.06 ± 0.01 (1.04–1.08). The mean WBL change per correction angle was 3.9 ± 0.3 (3.0~4.6)% in HBHTO and 4.1 ± 0.3 (3.1~4.7)% in OWHTO, which was significantly different in each of the 100 cases (*p* < 0.001).

The mean distance between the hinge point and center of the deformity was 31.60 ± 18.41 (12.6–99.1) mm in HBHTO (HBCD) and 47.96 ± 20.56 (20.4–118.8) mm in OWHTO (OWCD), which was significantly different in each of the 100 cases (*p* < 0.001). The mean difference in the distance from the hinge point to the CD between the HBHTO and OWHTO (dHCD) was 16.35 ± 4.74 (–2.68–23.91) mm (Table 3).

Correlation analysis revealed a strong positive correlation between CAD and HKA (r = 0.791, *p* < 0.001). All components of the HKA, including mLDFA (r = 0.261, *p* = 0.010), JLCA (r = 0.401, *p* < 0.001), MPTA (r = −0.342, *p* = 0.001), and leg length (r = −0.348, *p* = 0.001), were significantly correlated with CAD (Figure 4).

The OWCD (r = −0.505, *p* < 0.001), HBCD (r = −0.479, *p* < 0.001), and dHCD (r = −0.386, *p* < 0.001) were also significantly correlated with CAD (Table 4 and Figure 5).

Analysis of the components which comprise HKA yielded a multiple regression equation of
CAD = 3.953 + (−0.052 × MPTA) + (0.026 × JLCA) + (0.017 × mLDFA) + (0.010 × Width) + (−0.003 × OWCD) + (−0.002 × Leg length)

The standardized coefficient for each component was 0.173 for mLDFA, 0.234 for proximal tibial width, −0.305 for OWCD, –0.318 for LL, 0.344 for JLCA, and −0.495 for MPTA following an order of increasing influence on the CAD.

Comparison between different CD position revealed significant difference in HKA, JLCA, MPTA, CAD, OWCD, HBCD, and dHCD. In patients with high CD position, deformity parameters including HKA, JLCA, and MPTA were larger than in the low-CD group, and CAD was also larger in the high-CD group accordingly. Absolute distance of CD from the hinge was significantly further in the low-CD position group regardless of HBHTO and OWHTO. dHCD was larger in the low-CD position group (Table 5 and Figure 6).

## 4. Discussion

The most important finding of this research is that a larger correction angle is required in HBHTO to achieve an equal amount of WBL change with OWHTO, and the phenomenon is owing to the different hinge distance from the center of deformity, which can be translated into the concept of ‘Correction power of HTO.’

HTO is an effective joint-preserving surgical treatment option for patients with medial unicompartmental knee OA with varus malalignment. But, also, basically as a deformity correction surgery, HTO corrects the proximal tibia vara until it reaches the desired MPTA using the hinge point as the center of rotation of angulation (CORA). Therefore, different hinge positions in different HTO techniques would produce different correction powers. Considering the crossing point between the longitudinal mechanical axes of the femur and tibia as the CD, the hinge point of the HTO is located outside the CD medially (in OWHTO) or laterally (in CWHTO and HBHTO) at some vertical distance. Angular correction from the CORA outside the CD produces the intended angular correction with unintended translation, which inevitably causes lengthening or shortening of the OT site in HTO. However, the change in the length of the proximal tibia is not entirely added to or subtracted from the entire lower-limb length owing to the existence of the knee joint. A portion of the change in tibia length is converted to a power that influences knee joint alignment [10,11]. Owing to this unexpected alignment change, the correction power of a specific HTO technique was determined [12]. Similar to our interpretation, Bartholomeeusen et al. [13] reported that the knee joint line obliquity after HTO was not only the result of the size of angular correction at the hinge point, but rather a complex interaction between foot and knee position and size of correction. They mentioned that hip, knee, and ankle joint positions were all involved to adapt the angular and length change in the proximal tibia which was generated by HTO.

We applied the ‘Miniaci method’ [9] considering that this method will probably show us the difference in the required correction angle between two different HTO techniques to achieve the same amount of WBL change due to this method’s characteristics, which reflect different hinge points in the surgical design [14]. As we expected, there was a mean 0.78 ± 0.22 (0.4–1.5)° difference in the required correction angle between HBHTO and OWHTO to achieve WBL in 65% position (*p* < 0.001). For instance, in a patient with 7.7° HKA and 83.3° MPTA, a 12.0° correction was required in OWHTO, whereas a 12.7° correction was required in HBHTO to achieve 65% WBL. Moreover, the difference tended to increase with the increasing severity of deformity (Figure 3). We regarded this phenomenon as being associated with the difference in the hinge point distance from the CD. The mean hinge–CD distance was 31.60 mm in the HBHTO and 47.96 mm in OWHTO. The mean ratio of the hinge–CD distance in HBHTO relative to OWHTO was 63%. We divided the CD group according to its position and compared the radiological parameters between the two groups to further analyzed the influence of CD on CAD. Generally, CD was formed in the high position in patients with larger HKA and JLCA, and distance from the hinge point to the CD was shorter in the high-CD relative to the low-CD position group. The CAD in the high-CD group was significantly larger than in the low-CD group. What was interesting to us in this comparison was that increasing distance from the hinge point to the CD and dHCD according to the lowering CD position was inversely correlated to CAD. Correlation analysis of the parameters with CAD revealed their relationships more quantitatively. HKA and its components, including mLDFA, JLCA, MPTA, and leg length, were significantly correlated with CAD. The parameters related to increasing deformity, which included an increase in HKA, mLDFA, and JLCA, and a decrease in MPTA and LL, showed a positive correlation with CAD. The negative correlations of OWCD, HBCD, dHCD, and CAD implied that increasing the hinge–CD distance was related to a decreasing difference in the required correction angle between HBHTO and OWHTO (Table 4). However, these correlations between CAD and the hinge–CD distance parameters were difficult to understand because the results were paradoxical to what we supposed based on the CORA concept.

To the best of our knowledge, no study has compared the correction powers of HBHTO and OWHTO. Even studies comparing CWHTO and OWHTO in patients within a similar era have been rare. We believe that there was no temporal opportunity for such a comparison. Conventional CWHTO was an osteotomy technique used until the 1980s [1], while OWHTO has prevailed since the development of angular stable locking plates in the 1990s [5]. Surgeons do not have many opportunities to consider different techniques simultaneously for treating a single patient. However, the emergence of the HBHTO technique [6] and the revival of CWHTO [15,16,17,18,19] have made it possible for multiple osteotomy techniques to be considered in a single case, especially in cases requiring large angular correction. Ogino et al. [20] compared the correction powers of CWHTO and OWHTO. They reported that the mechanical axis change relative to the change in MPTA was similar between the two techniques, whereas there was a larger change in JLCA and MPTA in CWHTO. Further, they reported that the mechanical axis change relative to the change in the MPTA was significantly greater in OWHTO when assessed in preoperative planning, which was very similar to our study conditions and, per our results, HBHTO required a mean of 5.6% more angular correction at the hinge point than OWHTO in the WBL position, targeting 65%. Theoretically, 1°~2° more correction angle will be applied in HBHTO if the planned correction angle exceeds 20°. There were cases requiring an additional 1° correction in HBHTO compared to OWHTO at around a 15° correction angle in our series, and the patients who require those amounts of correction have approximately 9°–10° varus deformity. The influence of different hinge points on angular correction could ap-pear in the usual deformity range. Moreover, an additional 1–2° correction angle may increase the joint line obliquity to fall beyond the acceptable range and have a detrimental effect on the longer-term results of HBHTO. Considering the reported mean error rate of 6% when applying the planned correction angle to real surgical corrections [7], care should be taken to avoid overcorrection in HBHTO [10], especially in large angular corrections.

This study has some limitations. First, the number of included cases is relatively small for this kind of observational study. The results would have greater statistical power in larger cases. However, screening for potential HTO candidates using strict inclusion and exclusion criteria did not allow for a large case study. Second, this two-dimensional analysis of plain radiographic images did not consider sagittal plane factors of lower-extremity deformity. Further investigation using computed tomography is required. Third, the results in this study were not correlated to the postoperative results. Precise application of a surgical plan in HTO is usually limited owing to subtle errors during the surgical procedures. So, we decided not to correlate postoperative radiological results with the mathematical results in this study because postoperative results would not help in explaining this study. Possibly, the data in this study are mathematically correct only and only for the limited given calculated conditions. Finally, logical understanding and interpretation of the results are insufficient despite confirming some ‘paradoxical’ correlations between CAD and hinge–CD distance factors. Further research is warranted to elucidate the reasons behind this.

## 5. Conclusions

HBHTO required a 5.6% larger correction angle at the hinge point to achieve the same amount of alignment correction as OWHTO, which implicate lower correction power than OWHTO. The difference in correction angle between the two techniques tended to increase in patients with severe deformities or shorter legs. The distance from the hinge point and the center of the deformity were negatively correlated with the difference in the correction angle. Surgeons should be cautious to avoid habitually applying the same correction angle with OWHTO in performing HBHTO to avoid under-correction. The possibilities of generating unacceptably large joint line obliquity by applying HBHTO also should be considered.

## Figures and Tables

**Figure 1 diagnostics-14-01137-f001:**
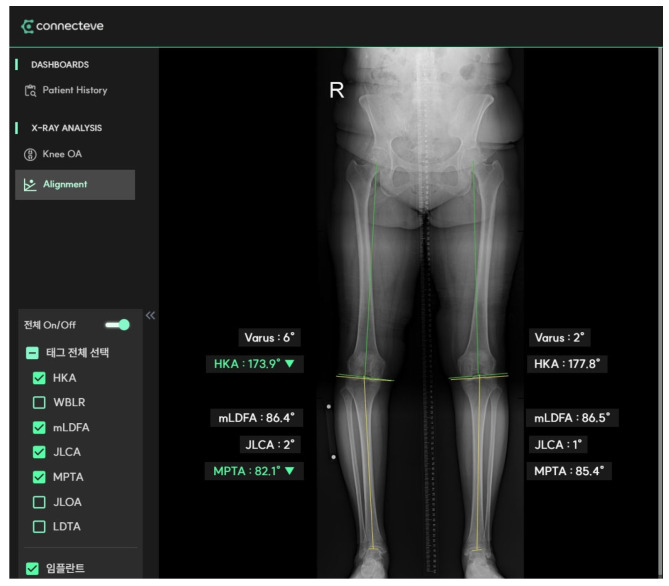
CONNECTEVE-X software was used to assess radiological parameters.

**Figure 2 diagnostics-14-01137-f002:**
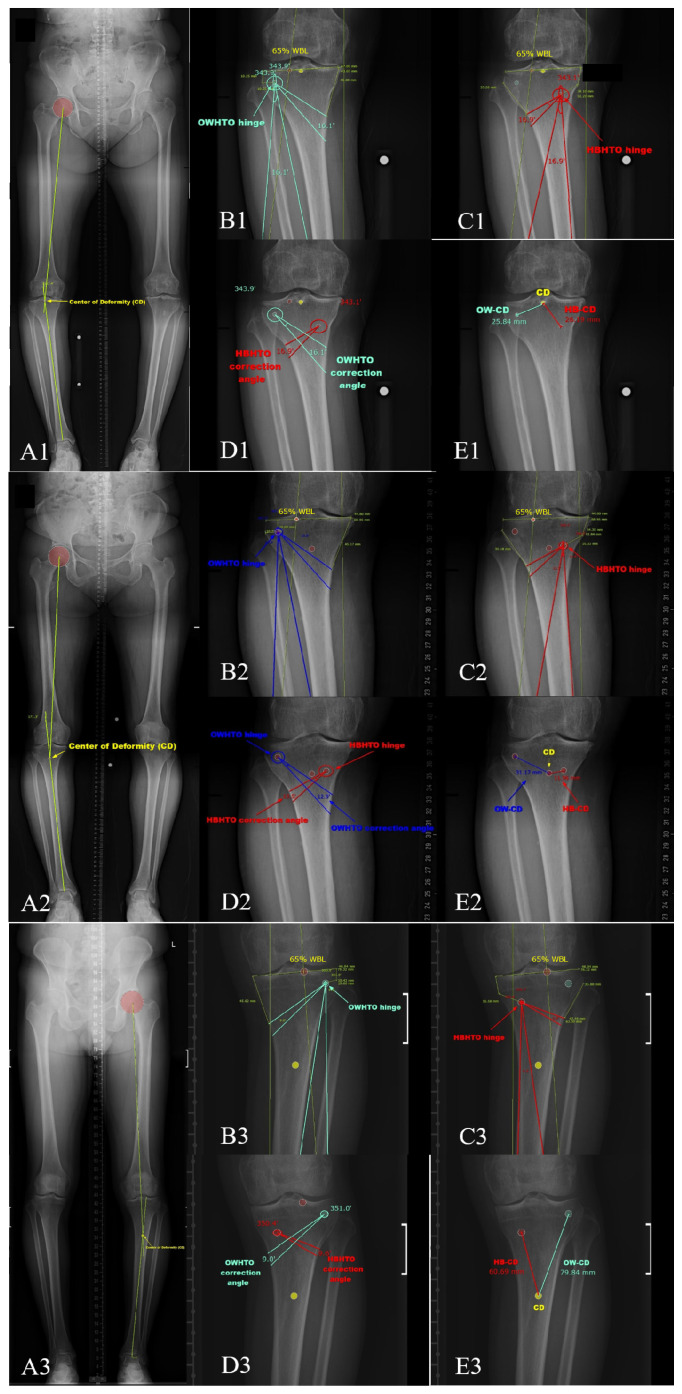
Center of deformity was designated as a crossing point between mechanical axis of the femur and tibia. Hinge points and correction angles were designated using Miniaci method. (**A**) Identification of center of deformity in the long standing radiograph (**B**) Designation of OWHTO plan based on the hinge point 10 mm distal to the lateral tibial plateau and 10 mm medial to the lateral proximal tibial cortex (**C**) Designation of HBHTO plan based on the hinge point located on the osteotomy line one-third from the medial tibial cortex (**D**) Relationship between center of deformity and two different hinge points (**E**) Distance from HBHTO hinge point to center of deformity was defined as HBCD and one from OWHTO hinge point to center of deformity was defined as OWCD; In case of highest CD position (**A1**–**E1**); high CD position(**A2**–**E2**); low CD position(**A3**–**E3**).

**Figure 3 diagnostics-14-01137-f003:**
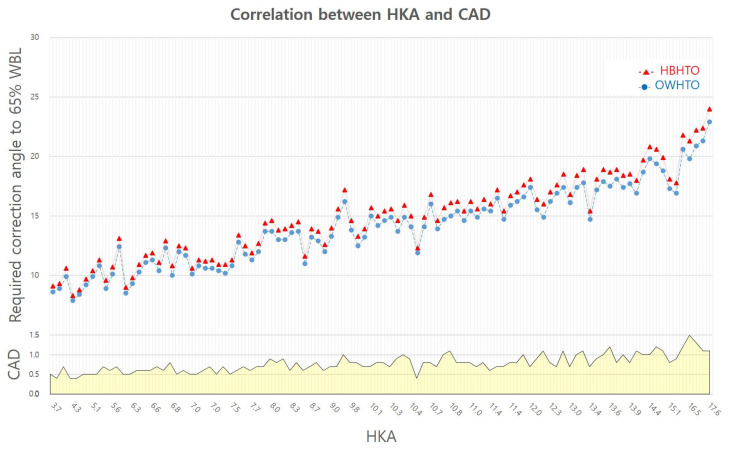
Difference in required correction angle tends to increase according to increasing hip–knee–ankle axis.

**Figure 4 diagnostics-14-01137-f004:**
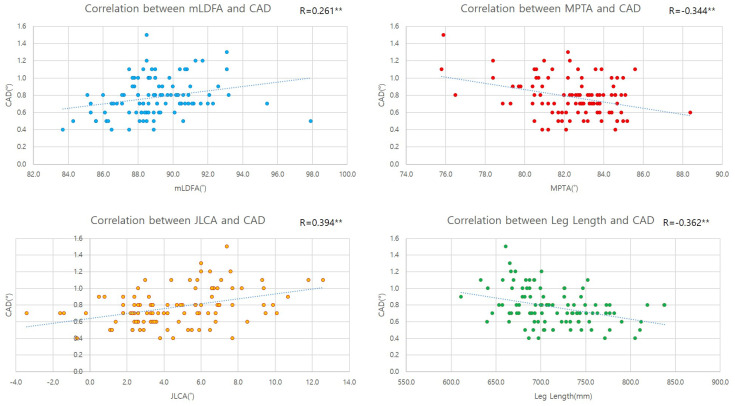
Correlation analysis between alignment parameters and CAD. **: The correlation is significant at the 0.01 level (both sides).

**Figure 5 diagnostics-14-01137-f005:**
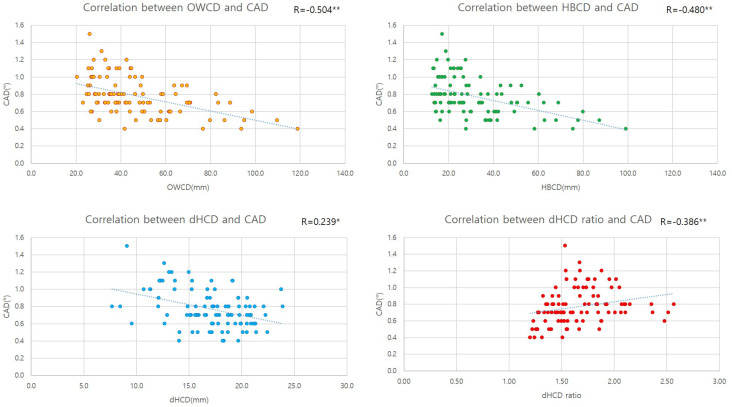
Correlation analysis between distance parameters and CAD. Correlation coefficient 0.000~0.250. *: The correlation is significant at the 0.05 level (both sides). **: The correlation is significant at the 0.01 level (both sides).

**Figure 6 diagnostics-14-01137-f006:**
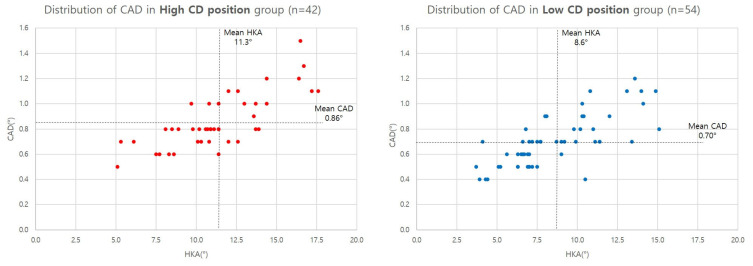
Distribution of CAD between different CD positions.

**Table 1 diagnostics-14-01137-t001:** Patient demographics and radiographic measurements.

Parameter (*n* = 100)	Value	Rater
Sex (M/F)	24/76	
Age (y.o.)	63.8 ± 10.3 (39~85)	
Side (Lt./Rt.)	53/47	
HKA (°)	9.94 ± 3.32 (3.7~17.6)	AI
mLDFA (°)	89.03 ± 2.23 (83.7~97.9)	AI
JLCA (°)	4.64 ± 2.93 (−3.4~12.6)	AI
MPTA (°)	82.49 ± 2.07 (75.8~88.4)	AI
Leg length (mm)	715.18 ± 46.05 (611.1~838.17)	Human
Length, tibia (mm)	309.91 ± 30.15 (103.8~378.2)	Human
Width, proximal tibia (mm)	72.88 ± 5.11 (64.58~86.38)	Human
WBL position (%)	7.4 ± 13.3 (−29.4~31.2)	Human

M: male; F: female; y.o.: years old; HKA: hip–knee–ankle axis; mLDFA: mechanical lateral distal femoral angle; JLCA: joint line convergence angle; MPTA: medial proximal tibial angle; WBL: weight bearing line.

**Table 2 diagnostics-14-01137-t002:** Intraclass correlation coefficient for the radiographic parameters.

	Intraobserver			Interobserver
	Observer 1	Observer 2	Observer 3	
Leg length (mm)	0.991	0.990	0.989	0.939
Length, tibia (mm)	0.985	0.979	0.958	0.964
Width, proximal tibia (mm)	0.921	0.935	0.932	0.928
WBL (%)	0.913	0.884	0.905	0.843
CD	0.957	0.882	0.920	0.872
OW hinge	0.917	0.955	0.891	0.858
HB hinge	0.854	0.904	0.883	0.818
Correction angle (OW)	0.922	0.857	0.881	0.850
Correction angle (HB)	0.931	0.903	0.894	0.888

WBL: weight bearing line; CD: center of deformity; OW: medial open-wedge high tibial osteotomy; HB: hybrid lateral closed-wedge high tibial osteotomy.

**Table 3 diagnostics-14-01137-t003:** Designated angles and calculations for 65% weight bearing line position in each HTO plan.

65% WBL Target	HBHTO Plan	OWHTO Plan	*p*-Value
Correction angle (°)	14.86 ± 3.47 (8.3~24)	14.09 ± 3.29 (7.9~22.9)	<0.001 *
CAD (°)	0.78 ± 0.22 (0.4~1.5)		
HB/OW angle ratio	1.06 ± 0.01 (1.03~1.08)		
WBL change (%) per correction angle	3.9 ± 0.3 (3.0~4.6)	4.1 ± 0.3 (3.1~4.7)	<0.001 *
Hinge–CD distance (mm)	31.60 ± 18.41 (12.6~99.1)	47.96 ± 20.56 (20.4~118.8)	<0.001 *
dHCD (mm)	16.35 ± 4.74 (−2.68~23.91)		
HBCD/OWCD ratio	0.63 ± 0.13 (0.39~1.11)		

CAD: difference in required correction angle between hybrid and open-wedge high tibial osteotomy plan; HB/OW; hybrid lateral closed-wedge high tibial osteotomy/medial open-wedge high tibial osteotomy; WBL: weight bearing line; dHCD: difference in the distance from the hinge to the center of the deformity between hybrid and open-wedge high tibial osteotomy plan; HBCD: distance between center of deformity and HBHTO hinge; OWCD: distance between center of deformity and OWHTO hinge. *: statistically significant difference.

**Table 4 diagnostics-14-01137-t004:** Correlation analysis for the radiographic parameters relative to CAD.

Relative to CAD	Pearson Correlation Coefficient (r)	*p*-Value
HKA	0.791 **	<0.001
mLDFA	0.261 *	0.010
JLCA	0.401 **	<0.001
MPTA	−0.342 **	0.001
Leg length	−0.348 **	0.001
Length, tibia	−0.131	0.202
Width, proximal tibia	−0.121	0.240
OWCD	−0.505 **	<0.001
HBCD	−0.479 **	<0.001
dHCD	−0.386 **	<0.001
HBCD/OWCD ratio	0.239 *	0.019

CAD: difference in required correction angle between hybrid and open-wedge high tibial osteotomy plan; HKA: hip–knee–ankle axis; mLDFA: mechanical lateral distal femoral angle; JLCA: joint line convergence angle; MPTA: medial proximal tibial angle; OW: medial open-wedge high tibial osteotomy; HB: hybrid lateral closed-wedge high tibial osteotomy; dHCD: difference in the distance from the hinge to the center of the deformity between hybrid and open-wedge high tibial osteotomy plan; HBCD: distance between center of deformity and HBHTO hinge; OWCD: distance between center of deformity and OWHTO hinge. *: The correlation is significant at the 0.05 level (both sides). **: The correlation is significant at the 0.01 level (both sides).

**Table 5 diagnostics-14-01137-t005:** Comparison of radiographic parameters between different CD position groups.

	High-CD Position (N = 42)	Low-CD Position (N = 54)	*p*-Value
HKA (°)	11.33 ± 3.07 (5.1~17.6)	8.60 ± 2.97 (3.7~15.1)	<0.001 *
mLDFA (°)	89.41 ± 1.55 (86.6~93.2)	88.72 ± 2.67 (83.7~97.9)	0.139
JLCA (°)	5.65 ± 2.81 (1.8~12.6)	3.63 ± 2.67 (−3.4~9.9)	0.001 *
MPTA (°)	83.09 ± 2.02 (75.9~88.4)	82.05 ± 2.07 (75.8~85)	0.016 *
Leg length (mm)	707.30 ± 46.34 (633.2~838.17)	722.23 ± 45.20 (611.1~818.82)	0.117
Length, tibia (mm)	307.63 ± 21.39 (272.1~378.2)	311.91 ± 36.08 (103.8~362.7)	0.471
Width, proximal tibia (mm)	72.47 ± 4.64 (64.58~84.84)	73.26 ± 5.63 (64.59~86.38)	0.457
Preop. WBL position (%)	1.9 ± 13.3 (−29.4~30.3)	12.6 ± 11.5 (14.1~31.2)	<0.001 *
WBL change (%)	63.1 ± 13.3 (34.7~94.4)	52.4 ± 11.5 (33.8~79.1)	<0.001 *
Correction in OWHTO (°)	15.73 ± 3.11 (8.9~22.9)	12.61 ± 2.80 (7.9~18.8)	<0.001 *
Correction in HBHTO (°)	16.58 ± 3.27 (9.6~24)	13.32 ± 2.97 (8.3~19.9)	<0.001 *
CAD (°)	0.86 ± 0.21 (0.5~1.5)	0.70 ± 0.20 (0.4~1.2)	<0.001 *
OWCD (mm)	33.17 ± 6.02 (24.87~49.88)	61.20 ± 19.15 (37.55~118.8)	<0.001 *
HBCD (mm)	17.67 ± 3.65 (12.6~26.87)	43.013 ± 18.18 (20.83~99.1)	<0.001 *
dHCD (mm)	15.51 ± 4.01 (7.67~23.91)	18.19 ± 2.43 (12.09~22.45)	<0.001 *

HKA: hip–knee–ankle axis; mLDFA: mechanical lateral distal femoral angle; JLCA: joint line convergence angle; MPTA: medial proximal tibial angle; OW: medial open-wedge high tibial osteotomy; HB: hybrid lateral closed-wedge high tibial osteotomy; CAD: difference in required correction angle between hybrid and open-wedge high tibial osteotomy plan; HBCD: distance between center of deformity and HBHTO hinge; OWCD: distance between center of deformity and OWHTO hinge; dHCD: difference in the distance from the hinge to the center of the deformity between hybrid and open-wedge high tibial osteotomy plans. *: statistically significant difference.

## Data Availability

The data presented in this study are available on request from the corresponding author.

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
