# Peer review of "Difference in Correction Power between Hybrid Lateral Closed-Wedge High Tibial Osteotomy and Medial Open-Wedge High Tibial Osteotomy was Associated with Severity of Varus Deformity and Different Hinge Distance from Center of Deformity"

_diagnostics, 2024, doi:10.3390/diagnostics14111137_

Round 1

Reviewer 1 Report

Comments and Suggestions for Authors

geometrical The paper presented for the review is aimed to investigate the potential difference in the required correction angle between HBHTO and OWHTO to achieve an equal amount of whole lower-extremity alignment correction.

The authors conducted the geometrical modelling study based on the full-length AP standing radiographs acquired from the patients before surgery. So, all the results are mathematical only. 

The wide spectrum of classical measurements was done including angular and metric data. Two variants of HTO were then modelled. Comparison of the 2 methods revealed the greater amount of the calculated angle needed for the same required correction in the cases hybrid closing wedge osteotomy.

Technically speaking the paper clearly answers the research question. But there are some concerns from the reviewer’s point of view which should be noticed and reflected in the discussion and/or limitations.

1. The authors used real long-leg radiographs of the patients before supposed surgery. The calculations were limited by angular measurements within proximal tibia only. No other modelled postoperative measurements were presented. First of all, I mean presumed joint line convergence angle. The simplified Miniaci method which was used by the authors is limited by tibial measurements only. The more advanced variant recommended by AO foundation (https://www.aofoundation.org/trauma/about-aotrauma/blog/2023_04-blog-open-wedge-high-tibia-osteotomy ) accounts the joint asymmetry represented by the joint convergence: self-correction to the joint line convergence angle (JLCA) should be subtracted from the angle.

2. The above mentioned point stresses the next concern - the authors postulated that the location of the center of the deformity (higher or lower) is the influencing factor. As the example they demonstrate the radiographs of 3 patients (fig 2) with patient 1 as an example of the high position of the CD. It’s clear the difference in the JLCA and the shapes of the proximal tibia between the first and the other cases. Logically and geometrically thinking, high (juxta-articular) position of CD is connected to the more proximal deformity (femur/joint) whereas the lower position - more related to tibia itself. And again, in the real world spontaneous correction of JLCA will interplay with the tibial correction which is not estimated neither mathematically nor clinically. 

3. The main finding (the difference between the needed correction angle for the 2 osteotomies) can partially or fully come from the fact that the lateral osteotomy is actually partial-wedge meanwhile medial - regular wedge-type. This purely geometrical difference also can play a role because the references point was linear, not actually angular (65% WBL).

4. Summarising, it is recommended to modify the paper - to stress throughout the text (and definitely in the aim and conclusion) that the data are mathematically correct only and only for the limited given calculated conditions. And the above mentioned points should also be clarified or listed among the limitations.

Author Response

  1. Thank you for your important question, reviewer. We wanted to reveal ‘How much change in the whole lower alignment will be induced by unit angular change in OWHTO and hybrid HTO?’ Our point was that simple. So, the other factors outside the tibia (mLDFA or JLCA) could be considered as influential factors, and correlation analysis was performed with them(line 191~196). JLCA itself is an angle formed between the proximal tibial and distal femoral joint line in plain radiograph, but its forming is influenced by body weight, collateral ligament competency, degree of articular cartilage wear, and medial meniscal status, etc, actually. So, we decided that JLCA could be analyzed as an influencing element of ‘correction power’ from the tibial hinge, but not a main role in our research. We think that influence of JLCA to whole lower limb alignment is another big subject, and requires separate major research. 
  2. Thank you. We also agree with your opinion that JLCA plays important role in lower extremity alignment and in the planning of alignment correction. So, we uniformly used ‘weight bearing’ long leg radiographs in all the cases, and tried to include the influence of JLCA in the calculation because we wanted the results to be realistic. We regard that the reduction in postoperative JLCA is in an area of uncertainty.
  1. Yes, we agree with your opinion. The location of hinge point in OWHTO is designated by the distance from the lateral tibial plateau and lateral tibial cortex, and it is constantly 10mm from both landmarks. But, in hybrid HTO, the location of hinge point is not constant and different from case to case because the point is designated by the ratio (2:1) in the osteotomy line.
  2. Thank you. We regard that it is a very important point to be stressed in our study. We mentioned it in line 319~325.

Reviewer 2 Report

Comments and Suggestions for Authors

Dear Authors,

            Thank you for submitting this interesting article.The discussion requires the following.

  1. The center of rotation for medial open wedge osteotomy is more lateral and is practically in the lateral cortex at the tip of the osteotomy.

   2. CORA more lateral would mean the calculations would be inappropriate.

    3. No data about what was achieved after surgery needs to be mentioned as a limitation.

      4. Mention about fibular osteotomy / fibular shortening / Nibbling of sup Tibio fibular joint in Lateral closed wedge osteotomy 

       5. Again the CORA is inappropriate in my opinion for sake of calculation. 

                                         Reviewer

Author Response

For question 1 and 2, Thank you very much. I also agree with your opinion that the real center of rotation in MOWHTO is located more lateral than hinge point and the calculation would be inappropriate. If we distract the osteotomy gap with the hinge as a center point, compressive deformation of bone bridge will occur somewhere between the lateral cortex and the hinge, and I agree that location may be called a real CORA. However, for now, we or anybody could not designate exact center of rotation beyond the hinge point even if they analyze CT scan of the surgical site. I regard that considering the hinge point as CORA is imperfect but rational choice for now because we plan and perform HTO up to that hinge point.

3. Thank you. We added the limitation in line no. 319~324. Because this study is regarding the accuracy of osteotomy in sub-1° difference, we considered that correlating real postoperative correction result with errors exceeding the sub-1° difference would not help empowering the logic of this study.

4. Thank you. We added the mention regarding fibular handling in line no. 99~100

5. Thank you. Please see the answer for comment no. 1 and 2.

Round 2

Reviewer 1 Report

Comments and Suggestions for Authors

The authors reworked the paper, according to the previous recommendation. The paper is recommended for the publication in the present version.

Reviewer 2 Report

Comments and Suggestions for Authors

Dear Authors

           Thank you for making the necessary corrections.

                                                                       Reviewer

Comments on the Quality of English Language

Minor editing of language is required.